# The Impact of the Need for Cognitive Closure on Attitudes toward Women as Managers and the Sequential Mediating Role of Belief in a Just World and Gender Essentialism

**DOI:** 10.3390/bs14030196

**Published:** 2024-02-29

**Authors:** Conrad Baldner, Antonio Pierro

**Affiliations:** Department of Socia and Developmental Psychology, Faculty of Medicine and Psychology, Sapienza University of Rome, 00185 Rome, Italy; antonio.pierro@uniroma1.it

**Keywords:** need for cognitive closure, attitudes toward women as managers, belief in a just world, gender essentialism

## Abstract

This research investigated the relation between the need for cognitive closure (i.e., a desire for epistemic certainty) and attitudes toward women as managers among men and women. In a cross-sectional study (total N = 241) collected in Italy, we found that need for cognitive closure, controlling for participants’ gender, was related to having more prejudice toward women leaders. Furthermore, the results revealed that the positive relation between the need for cognitive closure and negative attitudes toward women as managers was sequentially mediated by belief in a just world (i.e., the belief that people get what they deserve and deserve what they get and other people do not) and gender essentialism (i.e., the belief that women and men are distinctly, immutably, and naturally different, and thus have complementary skills to bring to the workplace). We suggest that men and women who are characterized by a need for cognitive closure are more sensitive to stereotypes of women as being incompatible with leadership roles. Either priming a low need for cognitive closure or providing contrary stereotypes could obviate the effect on beliefs in a just world and in gender essentialism that impedes progress towards greater gender equality in the workplace.

## 1. Introduction

How do people react when confronted with others who experience maltreatment? A simple answer is that people would react against this behavior. A more complex and accurate answer is that people would reject this behavior when they perceive it as unjust (e.g., discrimination) but accept it when it is instead perceived as just given the circumstances (e.g., punishment). Maltreatment, for one reason or another, is a common phenomenon, but often, people do not agree on whether it is just or unjust. In this work, we ask whether two systemic justification ideologies—belief in a just world [1,2] and gender essentialism [3]—mediate the previously studied relationship between the need for cognitive closure (NCC; i.e., the desire for stable and certain knowledge [4]) and negative attitudes towards women managers [5]. We will argue that the desire for stable and certain knowledge inherent in the NCC can prompt individuals to accept that people get what they deserve (i.e., a belief in a just world); this can prompt individuals to accept that we have fixed characteristics based on our genders (i.e., gender essentialism) and that guide us into gender-based roles; consequently, we can be more likely to mistreat women managers as a group that rejects these stereotypical roles.

Whenever individuals are faced with questions about the state of the world (e.g., does a specific group or person deserve mistreatment?) they open an epistemic process that closes when they arrive at an answer; the NCC helps explain how they navigate this process. As previously noted, individuals with an NCC desire and seek out any knowledge that provides perceived certainty and stability [4]. It has also been noted, since the earliest NCC research, that it is related to the acceptance of harmful stereotypes [6]. Given that specific cultural stereotypes represent sources of knowledge that apply to large groups of people and are resistant to change, they can be perceived to offer stability and certainty. 

According to a recent review [7], the issues women face in senior leadership roles, including compensation, promotability, and termination, have been explained by researchers in terms of role congruity theory [8]. Although this theory generally states that negative evaluations are a potential outcome of incongruity between the expectations of different roles held by the same person, it has been commonly applied to the case of women managers and leaders, who can be perceived as “too feminine” (and consequently insufficiently managerial) or “too managerial” (and consequently insufficiently feminine). Consequently, women managers can face issues in the workforce that are not shared by their men colleagues. Individuals in a given culture will be aware of the stereotypes within their culture, even if they do not accept them; we hypothesize that these stereotypes—in the current case, gender-based leadership stereotypes—will be more likely to be accepted by these individuals. 

Indeed, previous research has been conducted on this relationship, even after control-ling for participants’ gender and political orientations, in terms of both negative attitudes [5] (e.g., “It is not acceptable for women to assume leadership roles as often as men”) as well as preference for men in management roles [9]. This research has also identified hostile sexism [10] (i.e., negative attitudes towards nontraditional women) and the binding foundations [5] as mediators of this relationship, again after controlling for gender and political orientation. In light of the content of the binding foundation items [11] (e.g., “Whether or not someone conformed to the traditions of society”), this research has argued that they represent an affinity for traditional culture. In this sense, hostile sexism and, particularly, gender stereotypes and the binding foundations represent system justification ideologies in that they support the societal state of affairs (“the system”) even among individuals who are hurt by it [12,13].

Thirty years ago, Jost and Banaji [12] introduced the concept of system justification, or the acceptance and support of the current state of affairs, even if the individual, or their group, is harmed. This phenomenon can help explain the attitudes of members from disadvantaged groups that seem to violate their self-interest—for instance, women who support hiring practices that disproportionately favor men. Although there are self-report measures that directly assess system justification, Jost and Hunyady also identified a number of system justification ideologies, or those that guide individuals to support the current state of affairs, in addition to other construct-specific properties. For instance, they identified the need for closure as such an ideology. Even though individuals with this need have the desire for stable and certain knowledge, it can also lead them to system justification in that the current state of affairs can also be a source of stable and certain knowledge. Likewise, they identified factors analogous to Right-Wing Authoritarianism, Social Dominance Orientation, and the Belief in a Just World as system justification ideologies. Individuals who support authoritarianism [14] (RWA), perceive that the world is dangerous but can be tamed by social hierarchies [15] (SDO), or perceive that the world is basically fair and that people get what they deserve (i.e., Belief in a Just World, henceforth BJW) also tend to justify the current state with its consequent disadvantages (e.g., racism, sexism, income inequality). Any system justification ideology allows individuals to manage epistemic uncertainty—that is, they allow them to perceive the world as stable [16]. Although the two former factors—RWA and SDO—have received a fair amount of attention, De keersmaecker and Roets [17] have recently pointed out that the BJW has received less attention in the system justification literature. Their argument is justified, and the concept of the BJW seems inherently connected to system justification. For instance, in the original classic experiment, Lerner and Simmons [18] found that participants devalued a confederate, who received unnecessarily large shocks, when they were led to believe both that these were punishments for mistakes as well as that that they, the participants, were unable to stop the shocks. In this and similar studies, individuals had the need to believe that people got what they deserved as this allowed them to perceive the world as stable; of course, if people get what they deserve, then the system must be functioning well. 

De keersmaecker and Roets further argued that whereas ideologies like Right-Wing Authoritarianism and Social Dominance Orientation represent group-based system justification ideologies (i.e., protecting group norms), the BJW instead focuses on individuals [17], such as victims, and pointed to research that did not find a BJW effect on group-level discrimination [19,20]. Instead, they proposed that the BJW could be related to modern forms of prejudice, such as denial of discrimination and resentment towards disadvantaged groups who receive some form of help. 

In a superficial sense, we could expect that the BJW would thus predict negative attitudes towards women who are perceived to ask for special treatment (e.g., management hiring decisions) but would not predict negative attitudes towards women managers per se. However, if women managers are generally perceived to have unjustly obtained their positions, we would expect that women managers, as a group, would be met with disapproval and that this would be systematically linked with the BJW. On the other hand, we could instead expect that women who achieve management positions would be viewed positively under the BJW since they must have done something to deserve their high positions. For the BJW to be specifically associated with negative reactions, there is likely a mediator that represents both a belief about how world ought to be as well as a negative attitude towards nontraditional women. We identified gender essentialism [3] as a factor that satisfies both criteria. 

Essentialism, in general, is the idea that members of social groups share an essence that allows others to make useful inferences about its members [21,22,23]. Essentialism applied to gender is not only the idea that individuals share an informative essence based on their gender, but, in practice, also the idea that each gender has traditional—and natural—roles [3]. Men or women who do not fit into traditional roles can thus be perceived to violate nature. Gender essentialism has several interesting properties: not only is it a system justification ideology [24] but it also represents a perceived source of stable and certain knowledge. If someone wants to know whether, for instance, women are capable managers, they need only turn to these types of gender essentialist beliefs. Consequently, it is not surprising that essentialism, in general, is associated with the NCC [25]. 

To our knowledge, there has not yet been research on the BJW and gender essentialism; however, this relationship is plausible. It is clear that certain roles, like management, are typically held by men—even if this is slowly changing. More importantly, if specific roles are typically held by men or women, in a just world, they must be adapted to them. For this to be true, individuals must not only share a gender-based essence, but this essence must also be consistent with traditional gender roles. Consequently, the belief in a just world, which can be perceived to satisfy the NCC, could also result in the belief in gender essentialism. We could accordingly expect that people who seem to violate gender essentialism—such as women managers—would be at risk for maltreatment.

### The Present Research

Through a cross-sectional study conducted in Italy, the present research aimed to investigate the relation between the need for cognitive closure (i.e., a desire for epistemic certainty) and attitudes toward women as managers. We suggest that men and women who are characterized by a need for cognitive closure are more sensitive to stereotypes of women as being incompatible with leadership roles and therefore express more negative attitudes toward women as managers. Furthermore, we hypothesized that the positive relation between the need for cognitive closure and negative attitudes toward women as managers is sequentially mediated by the belief in a just world and, in turn, by gender essentialism. 

Our sample size was determined by a stop rule in which we decided to stop data collection after 10 days. However, our final sample size (241, see below) is also consistent with a moderate effect. According to the Monte Carlo Power Analysis for Indirect Effects application [26], a serial mediational model with moderate correlations between all variables results in a sample size of 244 at α = 0.05 and at 80% power. 

## 2. Materials and Methods

On a voluntary basis, two hundred and forty-one Italian participants (67.6% female; Mage = 27.96; SDage = 9.54; range: from 18 up to 69 years)—three participants fewer than our sample size estimate—completed the research questionnaire through Google Forms. (There was a significant difference in age across occupation (*p* = 0.031); currently enrolled students were younger than other occupation groups (currently enrolled students = 23.3; non-students in the workforce = 30.2; other = 42.5). There were no significant differences in age for gender (*p* = 0.703) or the gender-by-occupation interaction (*p* = 0.288)). Participants were recruited through direct social media platforms (e.g., Facebook) by research collaborators; in some cases, participants were personally acquainted with t. Interested participants were sent a link by research collaborators. A total of 46.9% of participants were currently enrolled students, 47.7% were non-students in the workforce, and 5.4% reported other status (e.g., retired, unemployed). Moreover, 2.1% had a middle-school education, 45.2% a high-school education, 51.9% reported being university graduates, and 0.8% had a Ph.D. Participants were recruited between May and July 2022. They were not compensated for their involvement.

Once participants gave their informed consent, they were requested to indicate their demographic information (i.e., age, gender, educational level, occupation). Subsequently, participants filled out a questionnaire aimed to collect the research measures of interest (as listed below). All the items were administered in Italian. In the ‘measures’ section, we provide example of those items translated into English. The last part of the questionnaire thanked and carefully debriefed participants.

Need for cognitive closure: Participants responded to the Revised Need for Closure Scale [27]. This is a brief 14-item self-report instrument designed to assess stable individual differences in the need for cognitive closure (e.g., “Any solution to a problem is better than remaining in a state of uncertainty”). Participants responded to these items on 6-point Likert scale ranging from 1 (Strongly Disagree) to 6 (Strongly Agree). A composite need for cognitive closure score was computed by summing across responses to each item. Previous research [27] has demonstrated that the revised version of NfCS has nomological validity and satisfactory reliability. In the present sample, reliability of the Rev NfCS was satisfactory as well (α = 0.76).

Attitudes Toward Women as Managers: Attitudes toward women in the role of a manager were measured with the Women As Managers Scale (WAMS) [28]. This scale consists of 21 items explaining the general (non)acceptance of women as managers and reflects different stereotypes of women holding managerial positions (e.g., “Women are not ambitious enough to be successful in the business world”). Respondents were asked to indicate the extent to which they agreed with each statement on a 7-point Likert scale, ranging from 1 (Strong Disagreement) to 7 (Strong Agreement). Support for the unidimensionality of the scale has been presented in different studies [29,30]. After reversing the positively worded items, ratings were summed across items to create an overall score for attitudes towards women managers (α = 0.80); higher scores indicated negative attitudes towards women managers.

Belief in a Just World (BJW): Participants responded to the Global Belief in a Just World Scale (GBJWS) [31]. The GBJWS, an appealing alternative to other BJW scales due to its relative brevity, unidimensionality, and high internal consistency [32], consists of a brief 7-item self-report instrument designed to assess differences in the belief of the existence of justice in the world (e.g., “I feel that people get what they deserve”; “I basically feel that the world is a fair place”). Participants responded to these items on 6-point Likert scales ranging from 1 (Strongly Disagree) to 6 (Strongly Agree). A composite BJW score was computed by averaging responses to each item (α = 0.85).

Gender Essentialism (GE): Gender essentialism was measured with the Gender Essentialism Scale (GES) [3]. This scale consists of 25 items that focus primarily on beliefs about and explanations for the nature of gender differences, including the belief that gender differences are discrete (e.g., “People tend to be either masculine or feminine; there’s not much middle ground”), biologically based and natural (e.g., “Differences between men and women are primarily determined by biology”), and fixed or inalterable (e.g., “Differences between boys and girls are fixed at birth”). Respondents were asked to indicate the extent to which they agreed with each statement on a 5-point Likert scale ranging from 1 (Strong Disagreement) to 5 (Strong Agreement). A composite gender essentialism score was computed by averaging responses to each item (α = 0.84).

Control variables: Age, gender (−1 = Male; 1 = Female), education, and social desirability were included as control variables. To control for socially desirable responding, we used the following two items: “I have never been late for an appointment or work” and “I have never hurt another person’s feelings”. Participants responded to these items on 6-point Likert scales ranging from 1 (Strongly Disagree) to 6 (Strongly Agree). The two items positively and significantly correlated with each other (r = 0.31, *p* < 0.001), and we averaged them in a social desirability score.

## 3. Results

Descriptive statistics and correlations between variables are presented in Table 1. NCC, BJW, GE, and gender were significantly correlated with (negative) attitudes towards women as managers: attitudes were more negative among men, as well as among participants with a higher NCC, with a higher BJW, and with a higher GE. NCC, BJW, and GE were also positively and significantly related. The belief that women and men are distinctly, immutably, and naturally different (i.e., GE) was significantly more pronounced among males.

We then tested the sequential mediating roles of BJW and GE in the relation between NCC and attitudes toward women managers. Gender, age, education, and social desirability were included in the model as covariates. The analysis was performed using SPSS PROCESS macro [33] Model 6. Ninety-five percent CIs were employed and 5000 bootstrapping resamples were run. The results obtained from the analysis are summarized in Table 2 and Figure 1. As can be seen, controlling for gender, education, age, and social desirability, the total effect of NCC on (negative) attitudes toward women as managers was significant and positive. Moreover, the direct NCC effect became non-significant when the mediators were included in the model, thus indicating that the effect of NCC on attitudes toward women managers was totally mediated by the mediators considered.

NCC was significantly and positively associated with BJW. BJW, for its part, positively and significantly predicted GE, which, in turn, significantly and positively predicted (negative) attitudes toward women as managers. Finally, and more importantly, the total effect and all specific NCC indirect effects through the mediators considered were significant (see Table 3). As can be noted, the specific indirect effect through BJW was significant (indirect effect = 0.027, 95% CI [0.0021, 0.0639]), as was the specific indirect effect through GE (indirect effect = 0.069, 95% CI [0.0251, 0.1244]). Importantly, also the indirect effect via, sequentially, BJW and GE was found to be significant (indirect effect = 0.011, 95% CI [0.0004, 0.0306]). Thus, as expected, the total indirect effect of NCC on attitudes toward women as managers via our mediators was found to be significant (indirect effect = 0.109, 95% CI [0.0512, 0.1730]).

Given the nature of our research, we further tested the moderating impact of gender on all the relations between our main variables, running a moderated mediation model (PROCESS Model 92 [33]). In addition to the paths estimated in the sequential mediation model described above, we regressed the following: (1) the attitudes towards women managers on the interaction between mean-centered NCC and gender, on the interaction between mean-centered BJW and gender, and on the interaction between mean-centered GE and gender; (2) the GE on the interaction between mean-centered NCC and gender and on the interaction between mean-centered BJW and gender; (3) and the BJW on the interaction between mean-centered NCC and gender. Age, education, and social desirability were included in the model as covariates. No significant moderating impact of participant gender on relationships between variables of interest was found.

## 4. Discussion

We hypothesized that the previously found relationship between the need for closure and negative attitudes towards women managers (i.e., a form of maltreatment of women) could be mediated by the belief in a just world and gender essentialism. Consequently, we can advance our knowledge on when individuals support, and engage in, the maltreatment of women managers. Future work can also ascertain whether this applies to other outgroups. Theoretically, individuals with a need for closure would be more likely to hold that the world is just as this can help satisfy their desire for stable and certain knowledge; if one wants to know whether some group deserves their (mal)treatment, they need only look at the current state of affairs. Those who believe that the world is just can also be more likely to support gender essentialism, which, in practice, is not just the idea that genders have essential characteristics but, further, that these characteristics support traditional gender roles. These beliefs can be appealing to those that also believe that the world is just as it should be clear both that men and women have historically had specific gender-based roles but also that these have continued into the present in modified forms. For instance, even though women are now frequently hired and promoted to management positions, they still face stereotypes based on these traditional roles. As a result, the need for closure, which superficially reflects the mere desire for stable and certain knowledge, can lead to a form of prejudice towards women through sources of knowledge—the belief in a just world and gender essentialism—which can be perceived to provide this type of knowledge.

Our results supported our hypothesis: the effect of the need for cognitive closure on negative attitudes towards women managers was serially mediated by belief in a just world and gender essentialism, controlling for gender, education, and social desirability. On the one hand, this research adds to our knowledge of the need for closure. Although the effect of the need for closure on attitudes towards women managers has been previously studied, these works assessed, as mediators, specific attitudes towards women (i.e., hostile sexism) [10] and general cultural standards (i.e., binding foundations) [5]. This is the first work that has studied the mediating role of ideas of how the world ought to be (i.e., belief in a just world). One the other hand, this research also supports recent work on the belief in a just world itself. De keersmaecker and Roets [17] recently argued that the belief in a just world has been given little attention as a system justification ideology, further arguing that, as opposed to system justification ideologies like Social Dominance Orientation and Right-Wing Authoritarianism, the belief in a just world instead represents an individual-focused system justification ideology. In other words, it should be associated with negative attitudes towards individuals who are perceived to demand, or benefit from, special favors. However, to this date, there has been little research that has continued this line of thought. Our research, however, is consistent with this point insofar as individuals who support traditional gender roles can perceive women managers as having benefited from such special favors. This conclusion also highlights the critical role of gender essentialism in that individuals who believe in a just world might also perceive that women managers have earned their position (i.e., women managers have gotten what they deserve). This interpretation only works if the belief in a just world is also associated with a factor, like gender essentialism, that supports the use of traditional gender roles.

Our research also had limitations that can be addressed in future research. One such limitation is the use of a cross-sectional self-report design. Future research could use longitudinal designs or otherwise could use experimental designs to assess the individual paths in the mediational model. These methods would provide evidence for the causal effect of the need for closure, belief in a just world, and gender essentialism. In other words, we could test the model across several studies: Study 1 could assess the effect of experimentally manipulated NCC on just-world beliefs, Study 2 could assess the effect of experimentally manipulated just-world beliefs on gender essentialism, and so on. Furthermore, our sample was primarily composed of women, which could be relevant given that we were interested in attitudes towards women. Even though a supplementary analysis included gender as an interaction term, future research could recruit either a sample composed of women or a larger sample that was designed a priori to detect interaction effects. Likewise, the measure of our dependent variable, the WAMS [28], was written in the 1970s; even if the thematic content of this scale is still relevant, a more recent scale could better capture modern attitudes towards women in management and leadership roles. Finally, we recruited a sample of Italian participants, and some researchers could be interested to find out whether these effects would apply to other participants from other nationalities. Although this is ultimately an empirical question for future research, in the current sample, we were able to replicate the basic association between the need for closure and negative attitudes towards women managers. We can otherwise see no reason why an Italian sample would inherently differ from those recruited in other Western countries.

In addition to future research that can address these limitations, there are a number of other avenues for research. For instance, we argued that gender essentialism, or similar constructs, is necessary in order for the belief in a just world to be negatively associated with negative attitudes towards women managers. Otherwise, the belief in a just world might very well be predictive of positive attitudes towards women managers. This possibility could be tested, for instance by including groups who score high and low on gender essentialism. Naturally, it would be of interest to assess whether this basic model could also help explain negative attitudes towards other outgroups. In this case, gender essentialism would need to be substituted for other, more relevant constructs. However, if our argument is valid, this new construct would need to justify traditional social roles. Finally, future research can assess this model in participants in the workforce and who have a woman supervisor. In place of the WAMS, a fairly general measure, we could assess attitudes towards their actual supervisor. Such a model could move this research into a more applied area.

## 5. Conclusions

In this research, we assessed the impact of the need for closure, or the desire for stable and certain knowledge, on a form of maltreatment of women, negative attitudes towards women managers. In particular, we found evidence that the belief in a just world and gender essentialism mediate this relationship, controlling for participants’ gender. Both men and women who are characterized by a need for cognitive closure can support knowledge sources that are perceived to offer stable and certain knowledge; in this research, we assessed the belief in a just world as one such factor. Individuals who believe that the world is just can also be more likely to support gender essentialism as essentialist, traditional gender roles can be perceived to offer natural evidence for the generally subordinate position of women in the workforce. As a result, negative attitudes towards women managers are more likely.

## Figures and Tables

**Figure 1 behavsci-14-00196-f001:**
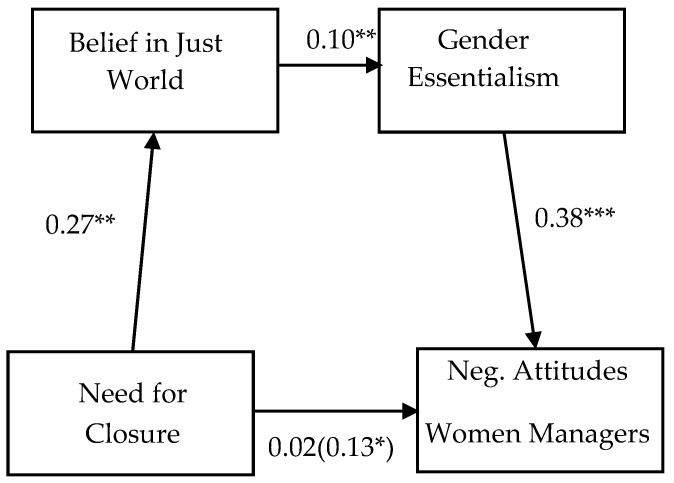
The indirect effect of Need for Closure on Negative Attitudes towards Women Managers through Beliefs in a Just World and Gender Essentialism. Total effect in parenthesis. Covariates are not shown for space considerations. * *p* < 0.05; ** *p* < 0.01; *** *p* < 0.001.

**Table 1 behavsci-14-00196-t001:** Correlations and descriptive statistics.

	1	2	3	4	5	6	7	8	M (SD)
1. Gender	-								-
2. Edu.	0.09	-							-
3. Age	0.03	0.09 †	-						27.9 (9.54)
4. SD	−0.03	0.02	0.14 *	-					3.05 (1.24)
5. NCC	−0.01	−0.07	0.23 ***	0.21 **	(0.76)				3.38 (0.67)
6. GE	−0.16 *	−0.10	0.20 **	0.09	0.30 ***	(0.84)			2.52 (0.55)
7. BJW	0.19 **	0.009	−0.05	0.23 ***	0.21 **	0.22 ***	(0.85)		2.48 (0.87)
8. WAMS	0.19 **	−0.09	0.01	0.14 *	0.17 **	0.41 ***	0.27 ***	(0.80)	1.74 (0.60)

Note: † *p* < 0.10; * *p* < 0.05; ** *p* < 0.01; *** *p* < 0.001; SD = Social Desirability; NCC = Need for Cognitive Closure; BJW = Belief in a Just World; GE = Gender Essentialism.

**Table 2 behavsci-14-00196-t002:** Mediation Model between NCC, Belief in a Just World, Gender Essentialism, and Negative Attitudes towards Women Managers.

	BJW			GE			WAMS		
	b	SE	*p*	b	SE	*p*	b	SE	*p*
Gender	−0.344	0.113	0.002	−0.147	0.072	0.043	−0.131	0.076	0.086
Age	−0.012	0.005	0.035	0.010	0.003	0.006	−0.003	0.003	0.311
Edu	0.078	0.096	0.414	−0.093	0.060	0.124	−0.040	0.063	0.522
SD	0.138	0.043	0.001	−0.006	0.028	0.812	0.035	0.029	0.230
NCC	0.270	0.083	0.001	0.182	0.053	<0.001	0.026	0.057	0.643
BJW	-	-	-	0.107	0.040	0.008	0.102	0.043	0.019
GE	-	-	-	-	-	-	0.383	0.068	<0.001

Note: SD = Social Desirability; NCC = Need for Cognitive Closure; BJW = Belief in a Just World; GE = Gender Essentialism.

**Table 3 behavsci-14-00196-t003:** Indirect effects.

	Indirect Effect	Bootstrapped SE	Bootstrapped 95% CI
NCC—BJW—WAMS	0.027	0.016	0.051, 0.173
NCC—GE—WAMS	0.069	0.025	0.025, 0.124
NCC—BJW—GE—WAMS	0.011	0.007	0.0004, 0.030

Note: NCC = Need for Closure; BJW = Belief in a Just World; GE = Gender Essentialism; WAMS = Negative attitudes towards Women Managers.

## Data Availability

The data are freely available from the corresponding author upon request.

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
