# Peer review of "The Impact of the Need for Cognitive Closure on Attitudes toward Women as Managers and the Sequential Mediating Role of Belief in a Just World and Gender Essentialism"

_behavsci, 2024, doi:10.3390/bs14030196_

Round 1

Reviewer 1 Report

Comments and Suggestions for Authors

After a thorough review of your manuscript "The Impact of the Need for Cognitive Closure on Attitudes Toward Women as Managers and the Sequential Mediating Role of Belief in a Just World and Gender Essentialism," I wish to congratulate you on a well-structured and meticulously researched paper. Your study provides valuable insight by analyzing the interplay between the need for cognitive closure and attitudes toward women as managers, especially through the mediation of belief in a just world and gender essentialism.

The way she has contextualized her research within a sound theoretical framework is remarkable. The references cited are pertinent and strengthen her arguments and conclusions. In addition, the clarity with which she has described the design, hypotheses, and methods of her research facilitates the understanding and replicability of the study.

Your arguments are coherent and convincing, and the discussion of the conclusions provides a fair balance between your findings and the existing literature. The presentation of results is clear and detailed, allowing readers to easily follow your reasoning and the implications of your findings.

In terms of improvements, although your study is rigorous, it could benefit from a broader discussion of limitations and possible future directions. This would not only provide a broader context for your research, but would also invite future studies to explore and expand on your findings.

In summary, their manuscript is a significant contribution to the field of study. His rigorous approach and detailed analysis are to be commended.

Author Response

  1. We were very pleased that you enjoyed the manuscript. We have added more info on the limitations and future directions, given the space limitations, both by adding more information on our existing limitations, adding new limitations, and by adding additional future directions.

Reviewer 2 Report

Comments and Suggestions for Authors

The research approach is interesting and original in the relationship between NCC and WAM. However, the text has some weaknesses that could be considered:

1.        The theoretical framework should include more depth in reference to the role congruity theory, stereotypes in leadership studies.

2.  Methodology:

a.        The sample is 240 interviews, but the text does not provide the size of the universe, who has been targeted by the google form?, how has it been contacted?

b.        Voluntariness means that the proportion of women is considerably higher and this may affect the explanatory power of the results. Even though it is random and the mean is taken into account.

c.         No data is provided on the distribution of respondents by age group. There may be a generational gender bias. Considering that half of the respondents are students.

3.        Results:

a.        There is a small inconsistency between table 3 and the paragraph explaining the data collected in the table.

b.        It would have been of interest to provide contrast data for the indirect effects. Comparison of the significant spill over effects could tell us which is statistically the most appropriate.

c.         Clearly, everything that passes through the relationship between GE and WAMS is correlated and significant. Either its direct relationship, 0.38, or its indirect relationship NCC-GE-WAMS or its indirect relationship with two intermediaries.

The approach is clear, the theoretical framework is limited, the sampling could be reinforced and the relationship between GE and WAMS influences the indirect approaches NCC, BJW, GE.

Author Response

  1. We have added recent literature on the application of role congruity theory to gender and leadership studies, as far as space limitations permitted.
    1. We included more information on our sample. Essentially, research collaborators (i.e., students who work in our lab) both directly contacted acquaintances and posted the Google Forms link on social media sites. We have also changed “students” to “currently enrolled students” and “workers” to “non students in the workforce”.
    2. This is possible, however we feel that it is ultimately a limitation of our recruitment method. Beyond controlling for gender, our only option is to recruit a new sample that is balanced for gender, which at the moment is unfortunately not possible. However, we have added this to our limitations.
    3. This is an interesting point. One on hand, we don’t have any way of creating non-arbitrary age groups (i.e., what is the start and and point for each age group?). However, we conducted one- and two-way ANOVAs with gender and occupation as the IV and age as the DV. If there were differences in age across age and gender they should be revealed in these analyses. We observed that were significant age differences across occupation, but that there were no effects for gender or gender x occupation. As these were not our main analyses we did not go into great detail explaining them in the manuscript, however we did briefly summarize the main points in a footnote when we introduce our sample.
    1. Thank you for pointing out this mistake; it has been corrected.
    2. It is not entirely clear to us which contrast effects we could have provided given that we tested a sequential mediation model. The most relevant information about the indirect effects that we can provide was displayed in Table 3.
    3. Indeed, each segment of the mediation model (e.g., NCC to BJW, BJW to GE, etc.) is significant. The best information we can provide about alternate models is displayed in Table 3.

Reviewer 3 Report

Comments and Suggestions for Authors

Dear Authors,

Thanks for the interesting article! The article’s findings are promising, but not entirely surprising when considering the obstacles to equal opportunities in the labor market and management positions. The research method is an online survey that collects people's opinions. When we are interested in opinions, the question arises whose opinions we are dealing with. This leads to the first point; I'm going to cover three points that I think authors need to consider when revising the manuscript.

First point. When the data was collected (e.g. a year, in ten days) and to whom it was advertised and where (e.g. university) (lines 144-146)? Half of the respondents were students and the other half were workers (lines 152-153). Were there differences of opinion between these groups? Students are supposed to represent the new generation with fresh thinking vs. workers in general. The data were collected online using Google Forms, so how can you be sure that the respondents are who they say they are? Were there rewards for participating? What was the reward?

Another point. The Women as Managers Scale (WaMS) is from 1974 and its items contain Western opinions from that era. Consider the following.

Item 8: “The business community should someday accept women in key managerial positions.”

Item 16: “Women are less capable of learning mathematical and mechanical skills than are men.”

Item 15: “On the average, a woman who stays at home all the time with her children is a better mother than a woman who works outside the home at least half time.”

These items are strange from today's perspective and should be revised to focus on management issues. Business has already accepted women for the highest management positions (e.g. Christine Lagarde). I wonder what mechanical skills have to do with managerial positions. During Covid-19, telecommuting at home became the standard for men and women working in management positions.

The manuscript has potential, but I encourage authors to revise WaMS to today's (Western) standards to make the argument stronger. Perhaps because of the 50-year-old scale, the framing of the article is uncritical, almost taken-for-granted: men and women. How does your approach adapt to managers with a transgender identity? France's new prime minister is openly gay. How do such social identities fit into the "old-fashioned categories" of men and women and the concept of "gender essentialism"?

The third point takes into account typos.

  • lines 34, 45, 168: check that the reference number is OK.

  • line 82: add at the end of the line: "henceforth BJW".

  • line 148: you have 241 respondents, why is the sample size 244?

  • line 225: insert under Table 1 the same note as in Table 2 + WAMS. 

  • Table 2. “WAM” -> “WAMS”; Insert “WAMS=....”; Remove “Gen=Gender” 

  • lines 232-233 and Table 3. The indirect effect is incorrect either in line 232 or Table 3. Also check that the numbers in the main text and Table 3 are identical (e.g. line 233: ".70" vs. Table 3: "0.69").

  • line 240 is missing ")".

Author Response

  1. We have added more information about how participants were recruited, including their compensation and the general time frame for participation. Although we cannot be sure if participants were honest in their responses, both for demographic and substantive questions, we doubt that this effected many participants for two reasons. First, participants did not have a clear motive to be dishonest, particularly regarding the demographic questions (e.g., why would participants lie about being students?). Second, our results were consistent with previous research in this area--general dishonesty in this study would only be likely if participants in several previous works, totaling thousands of participants, were not only dishonest but dishonest in the same direction. Regarding the differences between students and workers (or “currently enrolled students” and “non-students in the workforce,” as we refer to them in the revision), the differences in mean age are not great. Students are, on average, 23 years of age, as opposed to 30 for workers and 42 for the small number of “other” participants (this information was added in a footnote, originally in response to a comment from Reviewer 2). We don’t think that these are generational differences; in any case, these differences should be captured in the age variable. As we reported in the correlations table, age was significantly correlated with NCC and gender essentialism but not with BJW or WAMS. Indeed, the correlation between age and WAMS is almost exactly zero. As for any differences between students and workers, this is difficult to effectively assess in the current sample, beyond having included age as a covariate. In theory, we could add occupational status as a moderator, however the sample size was not based around a moderated mediated design and consequently this analysis would likely be underpowered. A possibility for future research, which we have added to the revised manuscript, would be to solely recruit participants who are in the workforce.
  2. This is an interesting point, and one that we had not considered prior to writing the manuscript. On one hand, completely revising the WAMS would itself require a lengthy research project which is not currently possible (i.e., in time to revise the manuscript). Beyond that, we wonder if the thematic content of the WAMS needs revising. To be sure, some items are old-fashioned—for instance, women in the highest positions are more common now than in the 70s--however we doubt that the thematic content is no longer relevant generally. We think that *anecdotal* evidence that women face discrimination in high positions is still abound (e.g., the problems Clinton faced in the 2016 US presidential election, see also the literature on the “glass cliff”). Beyond that, our opinion is that our results suggest that the WAMS is still relevant. Consider the information in Table 2: the WAMS had an average of 1.74; since it has a response scale from 1 to 7, on the surface this would suggest that it is no longer relevant. But also consider that it was correlated with the other variables, particularly with Gender Essentialism (r=.41). Gender Essentialism itself had a mean of about 2.5 on a response scale from 1 to 5, in other words the mean was only slightly below the scale midpoint. Rather than being thematically irrelevant, we think it is more likely that item content of the WAMS “turned off” some participants, even as participants were more likely to have higher WAMS scores as a function of NCC, BJW, and GE. As such, for the current revision, we added the development and use of a scale other than the WAMS as a future direction, however we would be surprised if results with such a scale would be very different from our current results.

  1. Thank you for pointing out these typos, they have been corrected.

Round 2

Reviewer 3 Report

Comments and Suggestions for Authors

Dear Authors,

The revised version looks good! I don't have anything more to add except to note that there's a typo (p. 4, line 162). I am looking forward to your future articles which might deal with a revised WAM.